# Mitochondrial Genomes from Fungal the Entomopathogenic *Moelleriella* Genus Reveals Evolutionary History, Intron Dynamics and Phylogeny

**DOI:** 10.3390/jof11020094

**Published:** 2025-01-24

**Authors:** Chengjie Xiong, Yongsheng Lin, Nemat O. Keyhani, Junya Shang, Yuchen Mao, Jiao Yang, Minghai Zheng, Lixia Yang, Huili Pu, Longbing Lin, Taichang Mu, Mengjia Zhu, Ziyi Wu, Zhenxing Qiu, Wen Xiong, Xiayu Guan, Junzhi Qiu

**Affiliations:** 1State Key Laboratory of Ecological Pest Control for Fujian and Taiwan Crops, College of Life Sciences, Fujian Agriculture and Forestry University, Fuzhou 350002, China; 18609200812@163.com (C.X.); linyongsheng0909@163.com (Y.L.); 15750811500@163.com (J.S.); maoyuchen301@163.com (Y.M.); jiaoyang0368@163.com (J.Y.); zhengmh0928@163.com (M.Z.); finy1015@163.com (L.Y.); hdpuhuili@163.com (H.P.); llb_77@126.com (L.L.); mutaichang@163.com (T.M.); zhumengjia_0529@163.com (M.Z.); 18105011103@163.com (Z.W.); 2Department of Biological Sciences, University of Illinois, Chicago, IL 60607, USA; keyhani@uic.edu; 3College of Literature and Law, Fuzhou Technology and Business University, Fuzhou 350715, China; qiuzhenxing2006@126.com; 4Forestry Diseases and Pests Control Station of Yongding District of Longyan City, Longyan 364000, China; yd5832216@163.com; 5College of Horticulture, Key Laboratory of Ministry of Education for Genetics, Breeding and Multiple Utilization of Crops, Fujian Agriculture and Forestry University, Fuzhou 350002, China

**Keywords:** *Moelleriella*, mitogenome, insect pathogen, phylogeny analysis, Hypocreales, Clavicipitaceae

## Abstract

Members of the genus *Moelleriella* (Hypocreales, Clavicipitaceae) are insect pathogens with specificity for scale insects and whiteflies. However, no mitochondrial genomes are available for these fungi. Here, we assembled seven mitogenomes from *M. zhongdongii*, *M. libera*, *M. raciborskii*, *M. gracilispora*, *M. oxystoma*, *Moelleriella* sp. CGMCC 3.18909, and *Moelleriella* sp. CGMCC 3.18913, which varied in size from 40.8 to 95.7 Kb. Synteny and codon usage bias was relatively conserved, with the mitochondrial gene arrangement being completely homologous to the gene order of 21 other species within the Hypocreales. Nevertheless, significant intron polymorphism was observed between *Moelleriella* species. Evolutionary analyses revealed that all 15 core protein–coding genes had ka/ks < 1, indicating purifying selection pressure. Sequence variation within the mitochondrial ATP synthase F0 subunit 6 (*atp6*) gene showed the largest genetic distance, with the ATP synthase F0 subunit 9 (*atp9*) gene showing the smallest. Comparative analyses of mitogenomes revealed that introns were the primary factor contributing to the size variation in *Moelleriella* and, more broadly, within Hypocreales mitogenomes. Phylogenetic analyses indicated that the seven *Moelleriella* species examined form a well–supported clade, most closely related to *Metarhizium*. These data present the first mitogenomes from *Moelleriella* and further advance research into the taxonomy, origin, evolution, and genomics of *Moelleriella*.

## 1. Introduction

The genus *Moelleriella* was established by Bresadola in 1896 to accommodate the type species *M. sulphurea* and belongs to Ascomycota, Sordariomycetes, Hypocreales, and Clavicipitaceae [1]. The genus *Moelleriella* is commonly parasitic on scale insects and whiteflies and currently comprises 65 species. The genus is characterized by brightly colored ascospores (mainly orange), pyriform to subglobose ascospores, cylindrical ascospores, filiform multiseptate ascospores that disarticulate at the septa within the ascus, and aschersonia–like anamorphs with fusoid conidia [2,3,4]. *Moelleriella* is predominantly found in the tropics, with several species isolated from the subtropics, and shows an Old World (OW)/New World (NW) disjunction in distribution [5,6]. In addition, geographically restricted species have been reported for some members of *Moelleriella* [7]. As sensitivity to ultraviolet light, solar radiation, and temperature are important factors known to affect the survival of entomopathogenic fungal spores, as well as of hosts such as whiteflies, most *Moelleriella*–infected specimens are found on the underside of forest foliage, i.e., leaves, and under conditions of high relative humidity [8,9,10]. Traditionally, *Moelleriella* species and members of closely related genera have been classified based on morphological differences, and, although the majority of *Moelleriella* species can apparently be cultured on standard media, a comprehensive substrate allowing for growth of all species has not been reported, and neither have fungal sexual structures been observed during growth on artificial media [11,12]. *Moelleriella* species can most often be reliably distinguished by observing the characteristics of spores, coupled to molecular taxonomic analyses of the sequences of various loci, e.g., *LSU*, *EF1–α*, and *RPB1*. The genus *Moelleriella* is currently divided into the “effuse” and “globose” branches based on DNA sequence data that correlated with stromatal morphology, with each branch containing 13 species [1,13]. However, the lack of more comprehensive molecular data and heterogeneity continue to be important obstacles in the study of phylogenetic analyses of *Moelleriella* species, as well as our understanding of their host specificity and distribution.

Scale insects (Coccidae and Lecaniidae, Homoptera) and whiteflies are significant agricultural pests and target a wide variety of host plants and crops. They act as important vectors for the transmission of a wide range of plant viruses and are often difficult to control. Species of *Moelleriella* have been recognized as pathogens of scale insects/whiteflies by modern naturalists in the late 19th century [14,15], with *M. libera* (anamorph *A. aleyrodis*) being one of the first species used for whitefly control in Eastern Europe and Asia [16,17]. Further studies have shown *Moelleriella* species to be promising for pest biological control, including as a resource for green agriculture development and as part of Integrated Pest Management (IPM) practices. However, a serious challenge is that most species of *Moelleriella* produce only a limited number of spores in artificial media, making large–scale production difficult [18]. Even with such current limitations, various metabolites derived from *Molleriella* species have been shown to possess a variety of activities of biopharmaceutical interest, including compounds that display anti–proliferative, anti–malarial, and anti–bacterial/fungal activities [19,20,21]. Knowledge concerning the genetic features of *Molleriella* species can, therefore, help in cultivation, bioprospecting, and pest control application.

Mitochondria contain their own genomes and are double–membrane organelles that provide energy for normal life activities in eukaryotes and are involved in signaling, genetic evolution, and other processes [22,23,24]. Genetic variation in the mitochondrial genome, due to its matrilineal inheritance properties, can be used to examine diversity and evolutionary processes [25,26,27,28]. Although the mitochondrial genomes of fungal genera vary significantly, most fungi typically contain 15 conserved protein–coding genes (PCGs), which include *atp6*, *atp8*, *atp9*, *cob*, *cox1*, *cox2*, *cox3*, *nad1*, *nad2*, *nad3*, *nad4*, *nad4L*, *nad5*, *nad6*, and *rps3* [29,30]. In addition to these conserved genes, two ribosomal RNAs, *rns* and *rnl*, and tRNAs also occur in fungal mitochondrial genomes [31,32].

Hypocreales comprises 303 genera and 14 families, and the genome sequences for a wide variety of these are currently available. However, mitogenomes are only available from fewer than 30 genera, with only three genera represented from the family Clavicipitaceae [33]. Additional mitogenomes would allow for greater depth in the analysis ofgenetic evolutionary processes within Hypocreales. To date, there are no reports on the mitogenomes of *Moelleriella* species, and here we provide for the frist time the sequences, assemblies, and annotations of seven *Moelleriella* species’ mitogenomes. These genomes were analyzed with respect to codon preference, intron dynamics, and gene homology, and extrapolated to evolutionary analyses of the core PCGs within Hypocreales. Furthermore, a phylogeny based on the mitogenome dataset was used to clarify the evolutionary position of *Moelleriella* in Hypocreales. Our results fill gaps in the mitochondrial genomic information of *Moelleriella* and provide reference data for molecular analyses on the origin, evolution, and diversity within *Moelleriella*.

## 2. Materials and Methods

### 2.1. Sample Collection and DNA Extraction

The specimens were collected from the provinces of Fujian, Jiangxi, and Zhejiang, and samples were kept dry after collection. One to two fresh ascospores were selected and dispersed as a spore suspension in sterile dH_2_O on an ultra–clean workbench and uniformly spread on the PDA plate. Isolates were single colony purified or allowed to grow as mycelium, after which the edge was used for subculturing. Once purified, growing mycelia were maintained in 30% glycerol and stored at −80 °C. Total DNA was extracted using the Fungal DNA Mini Kit (OMEGA–D3390, Feiyang Biological Engineering Corporation, Guangzhou, China). The concentration and purity of each sample were confirmed via spectroscopic analyses (Nanodrop, Thermo Fisher Scientific lnc., Waltham, MA, USA).

### 2.2. Mitogenome Sequencing and Assembly

High–quality genomic DNA was utilized for the construction of sequencing libraries. The libraries were sequenced using the Illumina HiSeq platform with a 2 × 150 double–end sequencing strategy. A total of 20.39 million (*M. zhongdongii*), 20.38 million (*M. libera*), 22.73 million (*M. raciborskii*), 25.84 million (*M. oxystoma*), 16.30 million (*M.* sp.C9), 20.78 million (*M.* sp.C3), and 21.27 million (*M. gracilispora*) raw reads were obtained. Raw sequences obtained were quality assessed and filtered to obtain clean sequences using the FastaP v0.20.0 [34], and we used Kraken2 v2.1.3 to identify mitochondrial sequences in the sequencing data [35]. Assembled filtered sequences were analyzed using SPAdes v3.14.1, and NOVO Plasty was used to validate the assembly of the mitogenome sequences [36,37].

### 2.3. Annotation of Mitogenomes

The annotation of protein–coding genes (PCGs), introns, rRNA genes, and tRNA genes of the mitogenome was conducted using the MFannot (https://megasun.bch.umontreal.ca/apps/mfannot/ (accessed on 2 August 2024)) and MITOS2 (https://usegalaxy.org/root?tool_id=toolshed.g2.bx.psu.edu%2Frepos%2Fiuc%2Fmitos2%2Fmitos2%2F2.1.3%20galaxy0 (accessed on 2 August 2024)) [38,39]; rRNA and tRNA sequence analyses were also examined using RNAweasel (https://megasun.bch.umontreal.ca/apps/rnaweasel/ (accessed on 2 August 2024)) and tRNAscan–SE v2.00 (https://www.psc.edu/resources/software/trnascan-se/ (accessed on 2 August 2024)) for validation [40]. Subsequently, the NCBI Open Reading Frame (ORF) Finder (https://www.ncbi.nlm.nih.gov/orffinder/ (accessed on 3 August 2024)) was employed to predict ORFs in the assembled mitogenomes, with analyses followed by functional annotation using Blastn and BlastP [41]. Intron–exon boundaries were detected using exonerate v2.2 [42]. All annotation results were manually corrected as needed. Graphical maps of the assembled mitogenomes were drawn using OGDraw v1.3.1 (https://chlorobox.mpimp-golm.mpg.de/OGDraw.html (accessed on 3 August 2024)) [43].

### 2.4. Sequence Analysis

The base compositions of the mitogenomes were calculated using the MEGA 11 v11.0.13 (https://megasoftware.net/ (accessed on 5 August 2024)). GC skew and AT skew were calculated according to the following formulas: AT skew = [A − T]/[A + T], and GC skew = [G − C]/[G + C] [44]. RSCU (Relative Synonymous Codon Usage) values were determined using condonW (https://sourceforge.net/projects/codonw/ (accessed on 5 August 2024)). A total of 15 core protein–coding genes were individually aligned using MAFFT v7.11 from 26 mitogenomes (7 determined here, and 19 from the NCBI database) [45]; the mitogenomes of these 19 species were selected based on high assembly quality representation of major clades. Nonsynonymous substitution rates (Kas) and synonymous substitution rates (Ks) of the core PCGs were calculated using the DnaSP v6 (http://www.ub.edu/dnasp/ (accessed on 5 August 2024)) [46]. The MEGA11 v11.0.13 software, using the Kimura–2–parameter (K2P) substitution model, was employed to detect paired genetic distances between the 15 core PCGs. BLASTn (e–value 10^−10^) was also used to compare the *Moelleriella* mitogenomes and to detect the presence of any large segments of intragenomic duplications within their genomes [47]. Tandem repeats were identified using the Tandem Repeats Finder (https://tandem.bu.edu/trf/submit_options (accessed on 6 August 2024)) [48].

### 2.5. Comparative Analyses of Mitogenomes and Intron Analysis

The length and compositional distribution of the mitogenomes of 26 species from the Hypocreales were examined using ggplot2 v.4.3.2 (https://github.com/tidyverse/ggplot2 (accessed on 7 August 2024)), and correlations between mitogenome size and the six components (core PCGs, RNA regions, intergenic regions, intronic regions, homing endonuclease genes (HEGs), and un_orf) were calculated. Intron dynamics analyses were performed as previously described [49]. To determine the insertion positions of introns, comparisons were conducted using MAFFT v.7.11, based on genetic code 4 and codons. The mitogenome from *M. gracilispora* was used as a reference to describe intron insertion positions. A total of three intron insertion position differences between species were still considered to have the same overall intron insertion distribution.

### 2.6. Mitochondrial Phylogenetic Analysis and Collinear Analysis

To determine the phylogenetic position of *Moelleriella* in Hypocreales, we analyzed the mitogenomes of various species and constructed a phylogenetic tree based on the amino acid sequences of fourteen conserved protein–coding genes in 70 Hypocreales species. Two Glomerellales species (*Colletotrichum aenigma* and *Colletotrichum gloeosporioides*) were employed as outgroups. Amino acid alignments were performed using MAFFT (v 7.11) with default parameters. Subsequently, the separate protein sequences were merged into combined datasets using PhyloSuite (v1.2.3) (https://github.com/dongzhang0725/PhyloSuite (accessed on 10 September 2024)) [50]. Bayesian inference (BI) analysis was conducted using MrBayes (v3.2.7) (http://nbisweden.github.io/MrBayes/, accessed on 10 September 2024), employing two runs of four chains and 2 × 10^6^ generations, along with a 100–sample frequency and 25% burn–in fraction. Iterations were considered to have reached convergence when the estimated sample size (ESS) exceeded 100 and the potential scale reduction factor (PSRF) approached 1. Maximum likelihood (ML) analysis was performed with IQ–TREE (v2.0.3) (http://www.iqtree.org/ (accessed on 10 September 2024)) under the LG + F + I + G4 model, incorporating 5000 bootstrap replicates and the Shimodair–Hasegawa–like approximate likelihood ratio test. The consensus tree was constructed using FigTreev.1.4.4 (http://tree.bio.ed.ac.uk/software/figtree/ (accessed on 20 September 2024)). Branches showing ML bootstrap support values (≥70) and Bayesian posterior probability (≥0.90) were considered significantly supported. The *Moelleriella* mitogenomes were compared using BLASTn (e–value 10^−6^), with homology analyses utilizing NGenomeSyn–1.41 (https://github.com/hewm2008/NGenomeSyn (accessed on 21 September 2024)) [51].

### 2.7. Data Availability

The complete mitogenomes of the seven *Moelleriella* species were deposited in the GenBank database under the accession numbers PQ367224–PQ367230.

## 3. Results

### 3.1. Characterization of the Seven Moelleriella Mitogenomes

The circular mitogenomes of seven *Moelleriella* species were determined as follows: 57,023 bp for *M. zhongdongii*, 40,823 bp for *M. libera*, 45,471 bp for *M. raciborskii*, 79,556 bp for *M. gracilispora*, 95,666 bp for *M. oxystoma*, 5389 bp for *Moelleriella* sp.C9 (*Moelleriella* sp. CGMCC3.18909, unpublished data), and 54,896 bp for *Moelleriella* sp.C3 (*Moelleriella* sp. CGMCC3.18913, unpublished data) (Figure 1). The highest GC content was observed in *M. raciborskii* (28.20%), while the lowest was seen in *Moelleriella* sp.C3 (26.10%). The average GC content was 27.10%. GC–skew values in the mitogenomes were from 0.0943 to 0.1128, and AT–skew values were from 0.0291 to 0.0572; the AT and GC skews were found to be positive for all seven mitogenomes. In total, the number of protein–coding genes present in the *Moelleriella* mitogenome determined ranged from 18 to 26 (Appendix A). Each mitogenome encoded 15 core PCGs (*atp6*, *atp8*, *atp9*, *cob*, *cox*1–3, *nad*1–6, *rps*3), two ribosomal RNAs (*rns* and *rnl*), and tRNA gene numbers ranging from 25 to 27 (Figure 1, Appendix A).

*Moelleriella oxystoma* was found to contain 12 uncharacterized open reading frames (un_ORFs), whereas *M. raciborskii* had only one, with other species ranging from 5 to 12. The number and distribution of introns exhibited significant variation. Intronic analyses revealed that *M. gracilispora* and *M. oxystoma* contained more introns than the other species examined and that these were mainly distributed in the *cob*, *cox1*, *cox3*, *nad1*, and *rnl* genes. *M. libera* had the fewest introns, which were mainly distributed in the *cob*, *cox*2–3, *nad1*, *nad5*, and *rnl* genes. Variable numbers of intronic ORFs encoding GIY–YIG and LAGLIDADG homing endonucleases were identified in the mitogenomes (Appendix A). In terms of sequence composition, PCGs consisted of the largest percentage (average = 34.78%), while RNA regions constituted an average of 11.33% of the total sequence (Appendix A).

### 3.2. Codon Usage Analysis

The codon usage patterns in different Hypocreales mitogenomes were analyzed. The *cox1* gene in 21 Hypocreales mitogenomes utilized ATG as its start codon, three others used TTG, and *Ophiocordyceps sobolifera* used TTA. Most core PCGs in the 26 Hypocreales mitogenomes used ATG as start codons, although the cob gene of *Stachybotrys chartarum* used GTG as start codons. In addition, the *cox2* gene of *Moelleriella* sp.C3, the *cox3* gene of *Stachybotrys chartarum*, and the *nad3* gene of *M. zhongdongii*, *M. oxystoma*, and *Moelleriella* sp.C9 used TTG as their start codon. Notably, *O. sobolifera* contained five distinct start codons, suggesting that species within this genus may possess unique or highly divergent translation initiation signals. Two different stop codons were found in the mitogenomes of the 26 Hypocreales species examined. These included TAA, which was detected in 15 genes, and TAG, found in seven genes (*atp9*, *cob*, *nad1*, *nad3*, *nad5*, *nad6*, *rps3*) (Appendix A). Codon usage analysis (Appendix A) indicated that the most commonly used codons in the *Moelleriella* mitogenomes were UUA (leucine; Leu), followed by AGA (arginine; Arg) (Figure 2A). Furthermore, codon usage bias in *Moelleriella* was essentially identical to other Hypocreales species (Figure 2B).

### 3.3. Repetitive Sequence Analysis

The *Moelleriella* mitogenomes analyzed contained 2–16 repeat regions. The size of these sequences varied from 35 to 405 bp, with the largest repeat identified in *M. oxystoma* in the *nad2* gene. The second largest repeat (284 bp) region was found in the *M. gracilispora cox2* gene. Pairwise nucleotide identities of the repeater sequences between the *Moelleriella* mitogenomes ranged from 76 to 100%, and intra–genomic repeats accounted for 1.02%, 0.32%, 0.38, 3.53%, 3.06%, 1.91%, and 0.37% of the mitogenomes of *M. zhongdongii*, *M. libera*, *M. raciborskii*, *M. gracilispora*, *M. oxystoma*, *Moelleriella* sp.C9, and *Moelleriella* sp.C3, respectively (Appendix A).

A total of 68 tandem repeats were found in the *Moelleriella* mitogenomes, with the number of repeats ranging from 4 to 14. The longest tandem repeat was found in *M. raciborskii* (136 bp). All tandem repeats were duplicated 1–2 times in the seven *Moelleriella* mitogenomes. The proportions of tandem repeat in *M. oxystoma*, *M. gracilispora*, *Moelleriella* sp.C3, *Moelleriella* sp.C9, *M. raciborskii*, *M. libera*, and *M. zhongdongii* were 0.29%, 0.51%, 0.74%, 0.77%, 0.81%, 1.22%, and 1.32% of the total mitogenome, respectively (Appendix A).

### 3.4. Genetic Distance and Evolutionary Rates of Core PCGs

Within the 15 core PCGs, the *atp6* gene had the highest median Kimura–2–parameter distance (K2P) genetic distance in the 7 *Moelleriella* species, followed by the *nad1* and *rps3* genes, indicating that these genes diverged early and have accumulated the highest level of change (mutations) as compared to the other mitochondrial PCGs. The *atp9* gene has the lowest median K2P distance, indicating the lowest degree of change (high conservation). These findings were further supported by analyses that showed that the *rps3* and *atp8* genes had the highest mean number of nonsynonymous substitutions per nonsynonymous site (Ka values) and the *atp9* gene the lowest. The *atp6* gene had the highest synonymous substitution rate (Ks), while the *atp8* gene had the lowest Ks value (Figure 3). The Ka/Ks values of the 15 core PCGs were all well below 1, indicating that these genes have undergone strong purifying selection during evolution.

### 3.5. Intron Dynamics in Moelleriella Mitogenomes

Intron numbers varied: *M. zhongdongii* (17), *M. libera* (9), *M. raciborskii* (13), *M. gracilispora* (34), *M. oxystoma* (33), *Moelleriella* sp.C9 (16), and *Moelleriella* sp.C3 (16), which were dispersed within 13 genes: *atp6*, *atp9*, *cob*, *cox1*, *cox2*, *cox3*, *nad1*, *nad2*, *nad4L*, *nad5*, *nad6*, *rnl*, and *rns* (Figure 4), with intron gain and loss seen. Introns were classified into intron position sets (IPSs) based on insertion positions in *M. gracilispora* genes, and common introns were defined as those occurring in the same IPS across the *Moelleriella* mitogenomes examined.

These data revealed a total of 138 IPSs (Figure 4A), of which 13 were common among the *Moelleriella* species, with each *Moelleriella* species also containing unique introns. The *cox1* gene had the highest number of introns, followed by the *rnl* and *cob* genes. In the *Moelleriella* species, most of the introns in the *cox1* gene were conserved, but only one intron in the *cob* gene was conserved (*cob*–393). Highly conserved introns were also found in the *nad1*, *cox2*, *cox3*, and *nad5* genes. Only *M. oxystoma* contained introns in the *nad6* and *nad2* genes, the presence of introns in the *rns* gene was only seen in *M. raciborskii* and *M. gracilispora*. Contrary to PCGs, non–coding genes (e.g., *rnl*) exhibited lower intron insertion position conservation (Figure 4A). Except for *M. raciborskii*, *M.libera*, and *M. zhongdongii*, the intron lengths of the remaining four *Moelleriella* species showed a clear unimodal distribution and *M. raciborskii* showed a bimodal distribution, whereas *M. libera* and *M. zhongdongii* showed a three–peaked distribution (Figure 4B). Intron lengths of all *Moelleriella* species were enriched in the 1000–2000 bp length region. Length distributions of common introns and species–specific introns showed no significant differences (Figure 4C). In addition, in most *Moelleriella* species examined, there were more introns in the mitogenome located between codons (phase 0) than between the first and second bases within codons (phase 1) or between the second and third bases (phase 2) (Figure 4D). The number of introns in phase 1 was comparable to that in phase 2, but, in *M. raciborskii* and *M. libera*, the number of introns in phase 1 was less than that in phase 2. *M. zhongdongii* showed a codon phase distribution with almost equal numbers in each category.

The conservativeness and variation of nucleic acid sequences (30 bp) around insertion sites of the same IPSs were also assessed (Figure 5). We found that the nucleic acid sequences around the same insertion sites were relatively conserved, with only a few base differences. Additionally, the insertion sites were almost identical, predominantly featuring bases, GT. This conservation suggests that the insertion sites are likely crucial for genomic function and/or stability, potentially related to gene expression regulation, intron processing mechanisms, and/or genomic integration.

### 3.6. Comparative Analysis of Hypocreales Mitogenomes

A comparative analysis of the mitogenomes of 26 species within the Hypocreales, including the seven *Moelleriella* species described herein was conducted. The sizes of the genomes exhibited considerable variability, ranging from 23,794 bp (*Orbiocrella petchii*) to 117,560 bp (*Ophiocordyceps lanpingensis*), with a mean value of 48,485 bp (Figure 6, Appendix A). Furthermore, the proportion of the five fractions (un_ORFs, intronic regions, intergenic regions, RNA regions, and core PCGs) varies between different species. Core PCGs and RNA regions account for more than 60% of the mitogenome in fungi, such as *Purpureocillium takamizusanense* and *Cordyceps militaris*, and only 1/3 or less in other species. The proportion of intergenic regions ranged from 3.22% to 28.01% across different species; meanwhile, un_ORFs percentages ranged from 0.50% to 20.02% across species. Conversely, in fungi, such as *Samsoniella hepiali*, *Cordyceps militaris*, and *Orbiocrella petchii*, there is no un_orf (Figure 6, Appendix A).

Furthermore, we investigated the potential correlation between mitogenome size and its content. Our findings indicated that significant positive correlations were found between mitogenomic lengths and HEGs (R^2^ = 0.865, *p* = 6.5 × 10^−8^), intronic regions (R^2^ = 0.986, *p* = 1 × 10^−23^), and un_ORF sequences (R^2^ = 0.699, *p* = 8.1× 10^−6^), which suggests that they exert a significant influence on the variability observed in mitogenomic size (Figure 7D–F). However, core PCGs and RNA region lengths displayed no notable correlation with mitogenomic size (R^2^ = 0.361, *p* = 0.0012, R^2^ = 0.05, *p* = 0.27) (Figure 7A,B). The observed differences in mitogenome composition across 26 Hypocreales species could be attributed to several factors, including repetitive sequence transfers and gain/loss events of introns and un_ORFs.

### 3.7. Gene Arrangement in the Mitogenome

We compared the arrangements of the 15 core PCGs and 2 rRNAs in the 26 Hypocreales mitogenomes and found that 22 members exhibited identical gene arrangements. However, isolates within *Stachybotrys*, *Metacordyceps*, *Clonostachys*, and *Acremonium* showed differences and displayed the following gene order: *cox1*, *nad1*, *nad4*, *atp8*, *atp6*, *rns*, *cox3*, *nad6*, *rnl*, *rps3*, *nad2*, *nad3*, *atp9*, *cox2*, *nad4L*, *nad5*, and *cob*. The gene orders in the mitogenomes of *Acremonium* and *Clonostachys* differed from the others only in the location of the gene *cox2*, which was found between the *nad4* and *atp8* genes, whereas *Stachybotrys* and *Metacordyceps* had significant differences in gene order, namely, as follows: *cox1*, *nad2*, *nad3*, *atp9*, *cox2*, *nad4L*, *nad5*, *cob*, *nad1*, *nad4*, *atp8*, *atp6*, *rns*, *cox3*, *nad6*, *rnl*, and *rps3* (Figure 8). In general, the mitogenomes of Hypocreales exhibited a high degree of conservation.

### 3.8. Phylogenetic and Synteny Analysis

To reconstruct the evolutionary history of the analyzed mitogenomes, phylogenetic trees were constructed using two inference methods, Bayesian inference (BI) and maximum likelihood inference (ML), based on the protein sequences of the 14 conserved mitochondrial protein–coding genes (PCGs) using a combined dataset from 70 different species (Appendix A). Identical tree topologies were obtained using the two phylogenetic inference methods, with *Colletotrichum aenigma* and *Colletotrichum gloeosporioides* used as the outgroup. All the major clades were well–supported in the phylogenetic tree (BPP (Bayesian Posterior Probability) ≥ 0.90; BS (Bootstrap Support) = 100), and those of the same genus grouped in the same branch and show similar or consistent characteristics in terms of GC content and gene conservation (Figure 9). This analysis also revealed a close evolutionary relationship between *Moelleriella* and *Epichloe*. To further assess the relationship between the mitogenomes of *Moelleriella*, homologous genes were compared using BLASTN. These analyses indicated a considerable number of homologous co–linear fragments within *Moelleriella* (Appendix A), with the longest between *Moelleriella* sp.C3 and *Moelleriella* sp.C9 (6737 bp). The highest number of homologous collinear fragments was observed in *Moelleriella* sp.C3 and *M. gracilispora*; *Moelleriella zhongdongii* had a lower level of homologous collinear fragments compared to other *Moelleriella* species, suggesting a more distant evolutionary relationship, consistent with the phylogenetic tree results. This may indicate that *Moelleriella zhongdongii* diverged earlier in the evolutionary process or that its genome has undergone more extensive changes, such as gene loss, gene rearrangements, or mutations. Additionally, to provide a more comprehensive depiction of orthologous relationships, we performed a comparative analysis using the complete mitogenomes of nine closely related species (seven *Moelleriella* species and two closely related species, namely, *Metarhizium album* and *Epichloe festucae*). We specifically focused on identifying and analyzing conserved synteny blocks exceeding 500 bp in length (Figure 10, Appendix A). The results indicate that, although there are few synteny blocks longer than 500 bp between the species, no gene inversions were observed, suggesting a high level of conservation of the mitogenomes.

## 4. Discussion

The mitogenome sizes of seven *Moelleriella* species ranged from 40,828 to 95,666 bp, with an average size of 61,038 bp. Previous studies have shown that the main factors contributing to variations in the size of fungal mitogenomes are varying numbers of introns, the accumulation and distribution of repetitive sequences, and the dynamics of intergenic regions [52]. Our data indicate that intron insertions were the main cause of mitogenome size variation between *Moelleriella* species and that the number of introns was proportional to genome size. More broadly, mitogenomes within the Hypocreales species varied considerably, ranging from 23,754 to 117,560 bp. Correlation analysis revealed that the sizes of the un_ORFs, HEGs, and intronic regions contributed to mitogenome length variation, which suggests significant intron drift and acquisition/loss, particularly of un_ORFs, in Hypocreales. Furthermore, GC skew and AT skew varied considerably among the 26 species examined, with most species having positive GC and AT skew, indicating someinter–genera differences.

As with almost all fungal mitogenomes characterized thus far, all seven *Moelleriella* species described herein contained a core set of PCGs that play an important role in cellular energy metabolism and functional maintenance [53]. Some differences in length, GC content, and base composition of the core PCGs were noted, and the evolutionary trajectory of these genes displays different levels of change and positive/purifying selection [54]. Codon usage is often linked to gene expression levels and can be used to gain evolutionary insight into species–relatedness. We found almost identical codon preferences in the mitogenomes of the seven *Moelleriella* species characterized, with some variation in the use of start and stop codons for the core PCGs. In addition, several non–conserved PCGs of unknown function were identified, suggesting unknown functions encoded therein and gene gain/loss in specific lineages. The Ka/Ks ratio is commonly used to estimate the selective pressure experienced by specific genome PCGs during evolution and reveal the underlying genetic mechanism. In our comparison of 14 PCGs in seven *Moelleriella* mitogenomes, all PCGs were negatively selected during the evolutionary process, implying that these genes are highly conserved. In particular, significantly lower Ka/Ks values were observed for five genes (*atp9*, *cob*, *cox1*, *cox2*, *nad4L*) in the present investigation, indicating that intense purification acted on these genes in maintaining their indispensable functions in mitogenomes.

Introns are prevalent in fungal mitogenomes and are mainly divided into groups I and II, with the *cox1* gene usually containing the largest number of introns, and intron polymorphisms affect the size and evolution of fungal mitotic genomes [55,56,57]. IPS analyses were used to delineate and compare the location of intron insertions. These analyses revealed that intron insertion sites exhibit frequent gain/loss dynamics, and none of the introns were shared by all seven *Moelleriella* species examined. These data are similar to what has been observed in *Cordyceps* and other fungi [58]. Approximately 5–10% of group I introns typically carry HEGs that can potentially act as mobile genetic elements [59]. A total of 138 introns were identified in our *Moelleriella* analysis, most of which were group I with HEGs. While it is unclear which may be “active”, these findings may account for the diversity of intron insertion positions found. In addition, introns were evenly distributed in the *Moelleriella* mitogenomes, and intron positions in different species were most prevalent between codons (“phase 0”), similar to what has been commonly described [60]. Overall, the fact that the *Moelleriella* species examined share a small number of intron insertion positions suggests the loss of ancestral introns and large changes in intron insertion positions during evolution. We further assessed the conservation of the nucleotide sequences around identical IPSs in the *Moelleriella* species and found that the insertion sites displayed certain base preferences.

Compared to nuclear genomes, mitogenomes typically evolve at faster rates and are more prone to accumulating mutations, which can increase their analytic utility for monitoring evolutionary changes during species differentiation. One aspect of this is the order of mitochondrial genes, which can undergo rearrangement, which can readily be seen in genome alignment analyses, and hence has been used as an important reference for reconstructing the evolutionary relationships [61,62]. However, mechanisms of rearrangement of fungal mitogenomes may be more intricate than previously thought, and some caution should be taken in deriving conclusions [63]. Despite gene rearrangements being widespread in fungi, our study found that the *Moelleriella* mitogenomes have a consistent gene arrangement which is generally conserved throughout the Hypocreales. In *Clonostachys* and *Acremonium*, changes in gene arrangement were observed with respect to the *cox2* gene, whereas *Stachybotrys* and *Metacordyceps* had unique mitochondrial gene arrangements, consistent with previous findings [64]. Recent studies have shown that the mitogenomes of Hypocreales species display six different patterns of gene arrangement and five different patterns of *nad2*/*nad3* connectivity, most of which are rare, and that the rare patterns are restricted to a few fungal species or groups of fungi. Our data do show, however, important rearrangements in the *Moelleriella* mitogenomes, consisting of mostly small homologous fragments, with many homologous co–linear genomic blocks shared with close relatives, while the 15 core PCGs and 2 rRNA genes were completely conserved. These data suggest *Moelleriella* species may have undergone important changes during evolution centered around mechanisms that cause frequent short fragment rearrangements, with certain regions more prone to recombination (“hot spots”), especially those near repetitive sequences or gene spacer regions. It is well recognized that the mitogenome is independent of the nuclear genome and has characteristics of more rapid evolution. Mitochondrial genes have been used as reference “molecular markers”, as tools for studying the phylogenetic placement and evolution of fungi [65]. In our comprehensive comparative analysis involving *Moelleriella* with *Metarhizium album* and *Epichloe festucae*, no inversion regions were observed. This suggests that Hypocreales species may have undergone lineage–specific genomic rearrangements in their mitochondria, leading to a scarcity of long synteny blocks. Rapid genomic evolution is typically associated with such genome rearrangements, whose signatures result in shortened syntenic regions.

*Moelleriella* species have important potential for insect biological control, especially against whitefly and scale insect pests [1]. Our data and phylogenetic analysis of *Moelleriella*, using the mitochondrial genome, can allow for the accurate classification of this genus. Based on the combined mitochondrial gene set and two phylogenetic inference methods, we obtained a strongly supported phylogenetic analysis of 68 related species within the Hypocreales. Our results indicate strong support for the seven identified *Moelleriella* species clustered together within a distinct clade with a high level of support, with *Epichloe* being the closest relative to *Moelleriella*. This demonstrates that the mitochondrial gene is a reliable molecular marker for the analysis of phylogenetic relationships of Hypocreales and perhaps other fungal groups. More mitogenomes are, however, needed to comprehensively assess the origin and evolution of the Ascomycota.

## Figures and Tables

**Figure 1 jof-11-00094-f001:**
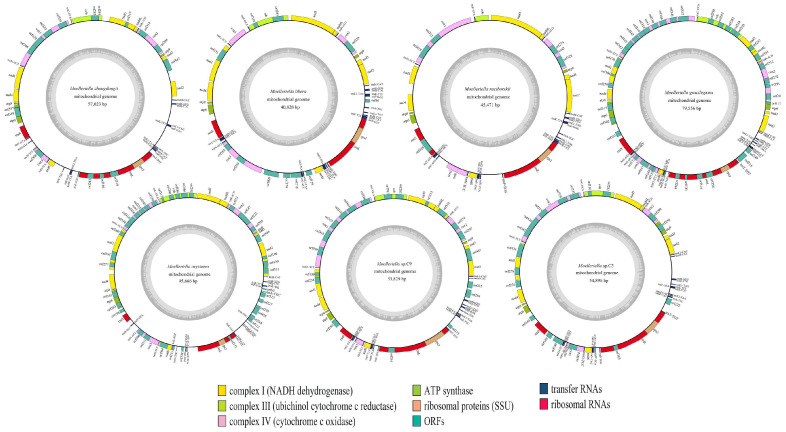
Circular maps of the mitogenomes of seven *Moelleriella* species. Genes are represented by different colored blocks. All functional genes were in the same strand. The gray plot in the inner circle indicates the GC content.

**Figure 2 jof-11-00094-f002:**
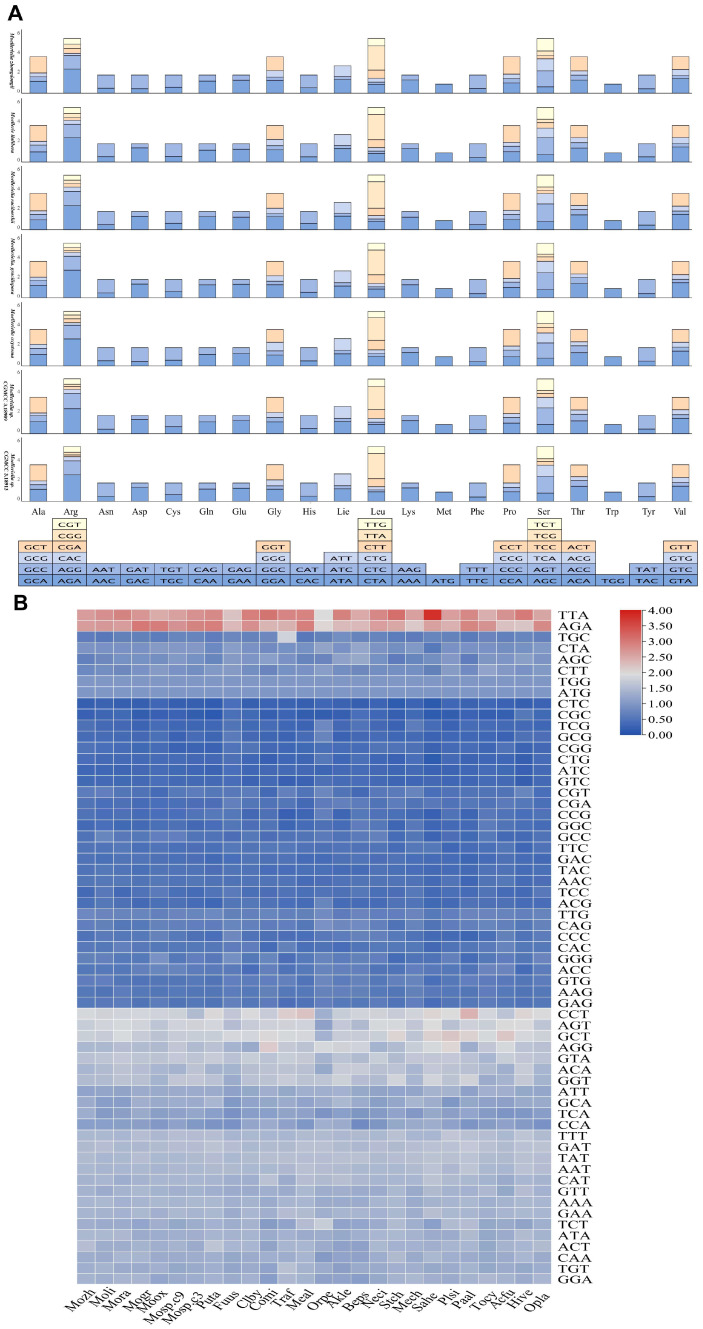
(**A**) Stacked column plots of the RSCU of the seven *Moelleriella* mitogenomes. (**B**) Heatmap of the RSCU of the mitogenomes of 26 Hypocreales species.

**Figure 3 jof-11-00094-f003:**
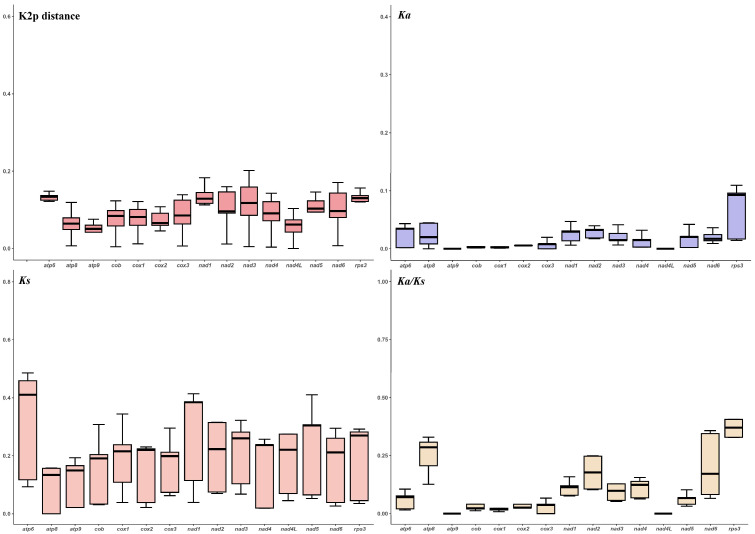
Genetic analysis of the 15 protein–coding genes (PCGs) conserved in the seven *Moelleriella* mitogenomes. K2P, the Kimura–2–parameter distance; Ka, the mean number of nonsynonymous substitutions per nonsynonymous site; Ks, the mean number of synonymous substitutions per synonymous site.

**Figure 4 jof-11-00094-f004:**
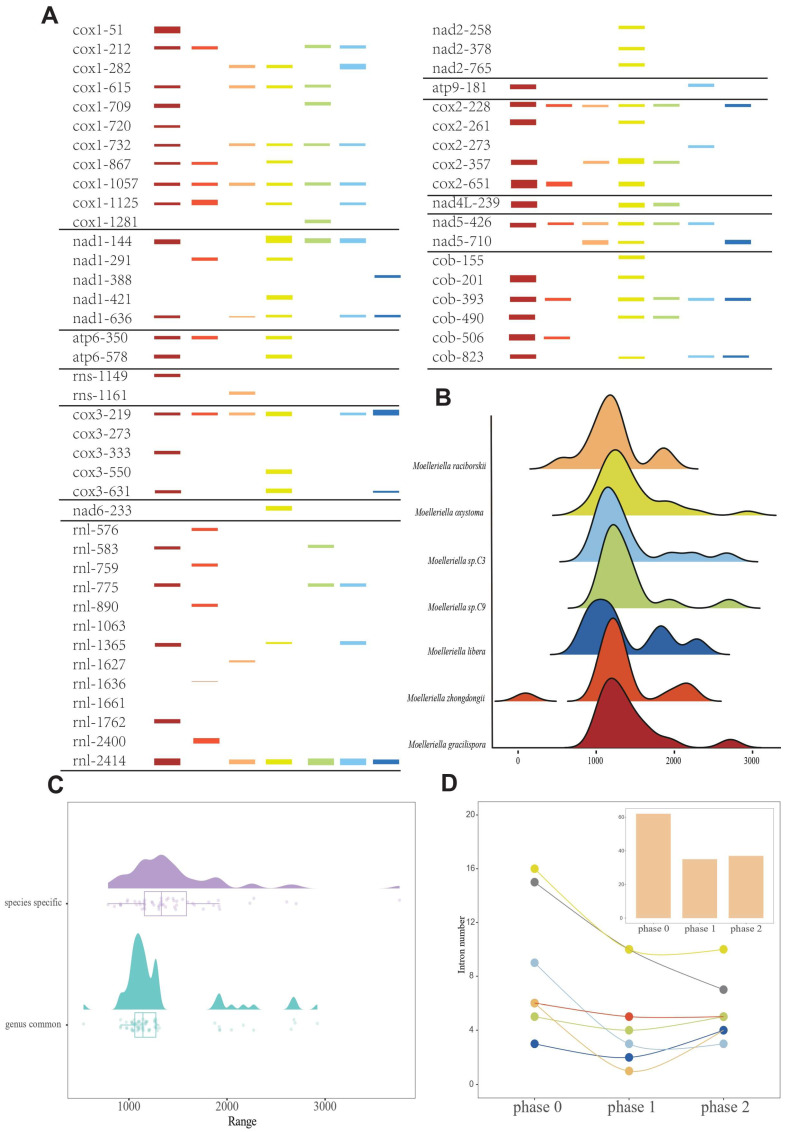
Intron position sets (IPSs) in mitogenomes of the genus *Moelleriella* and characteristics of the introns of *Moelleriella* species. (**A**) Bar graph indicating intron dynamics among seven *Moelleriella* species. Intron insertion positions were referred to coding sequences of the *M. gracilispora* genes. The height of each bar corresponded to its respective intron length. (**B**) The length distribution of the total introns of the *Moelleriella* species. (**C**) Box plots indicate length distributions of common introns and species–specific introns, respectively. (**D**) Codon phase distributions of intron insertion position.

**Figure 5 jof-11-00094-f005:**
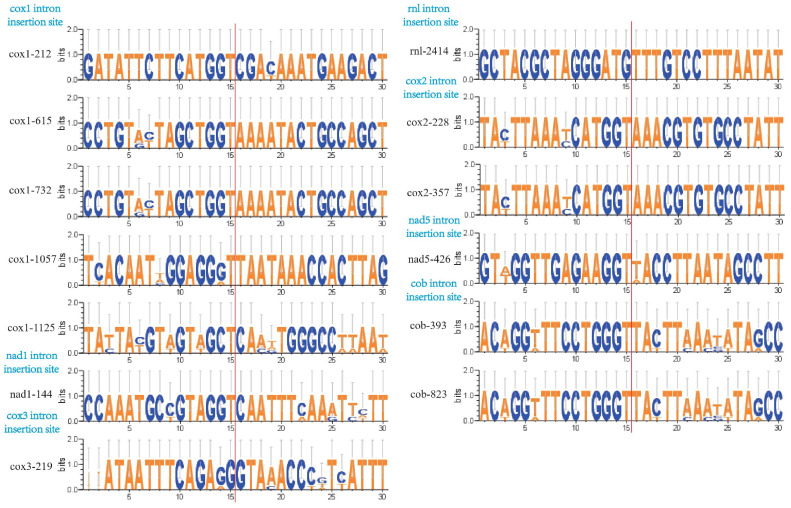
Consensus sequence maps of the insertion sites and surrounding sequences of orthologous IPSs in the coding regions of *Moelleriella* (−15 bp–15 bp).

**Figure 6 jof-11-00094-f006:**
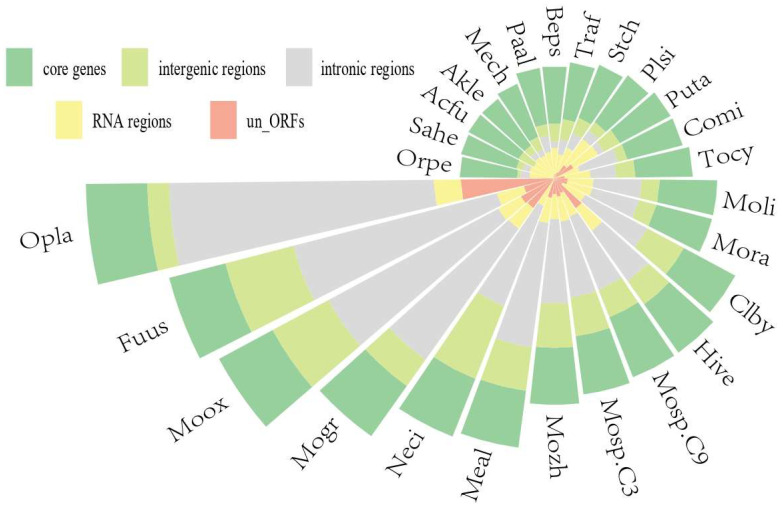
The lengths of RNA region, core PCGs, intronic region, un_ORFs, and intergenic region in 26 Hypocreales species.

**Figure 7 jof-11-00094-f007:**
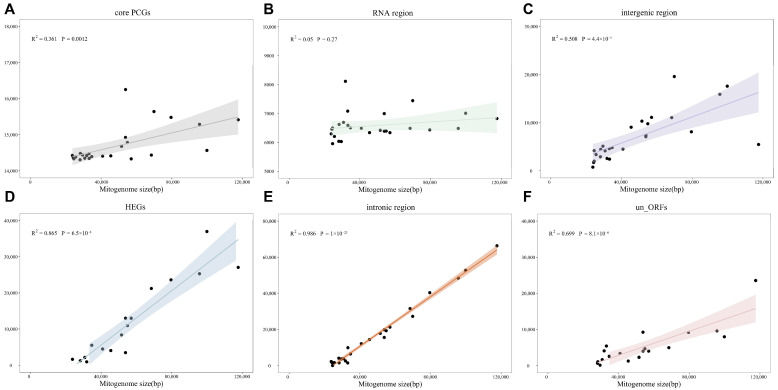
Correlation analysis between the lengths of mitogenomes and the lengths of (**A**) core PCGs, (**B**) intergenic region, (**C**) HEGs, (**D**) RNA region, (**E**) intronic region, and (**F**) un_orf.

**Figure 8 jof-11-00094-f008:**
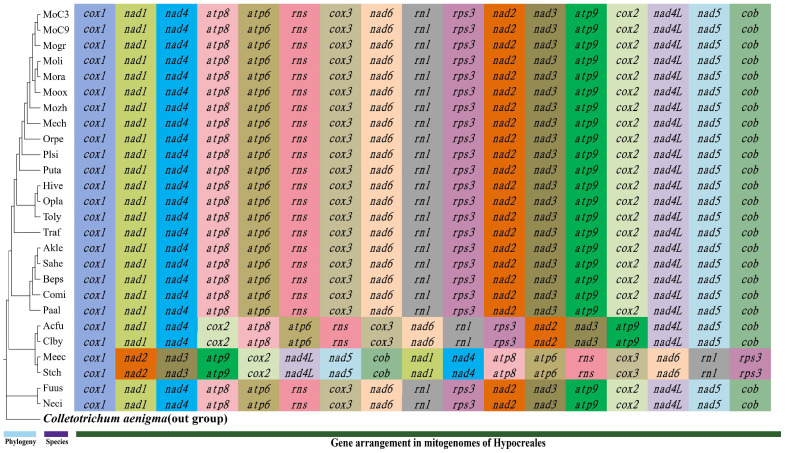
Gene order analyses for 26 Hypocreales mitogenomes. The same gene is represented by the same background color. All genes (fifteen core PCGs and two ribosomal RNAs) are shown in the order of their appearance in the mitogenome, starting with *cox1*.

**Figure 9 jof-11-00094-f009:**
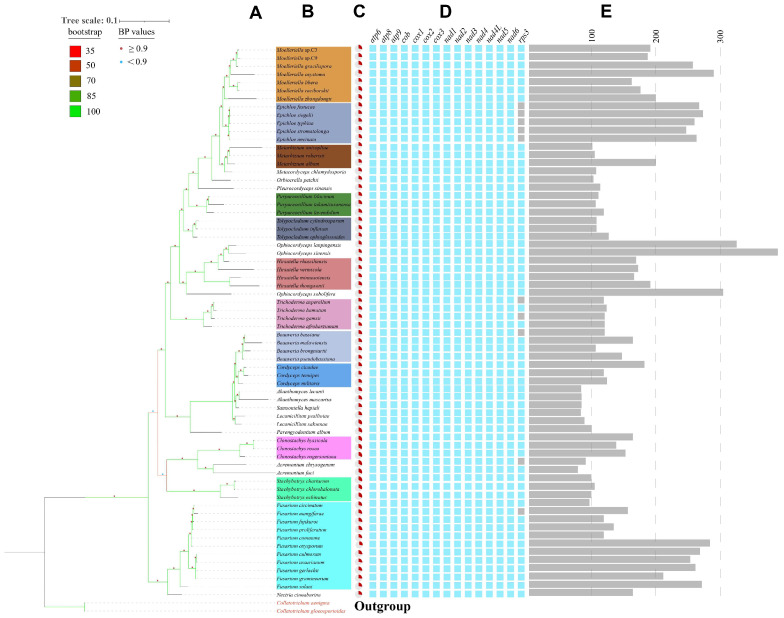
A phylogenetic tree inferred from concatenated mitogenomic PCGs of 70 species, based on Bayesian inference (BI) and Maximum likelihood (ML) methods. (**A**,**B**) Tree topology and species list showing the branching of lineages. The bootstrap values of tree nodes were color–coded. Nodes marked with red and blue points indicated BPP values equal to [0.9, 1] and <0.9, respectively. Genera containing ≥ 3 species were filled with the same colors in the species list. (**C**) GC contents. (**D**) The colored blocks represents the presence of fungal standard PCGs in the related species. (**E**) Genome sizes of corresponding mitogenomes.

**Figure 10 jof-11-00094-f010:**
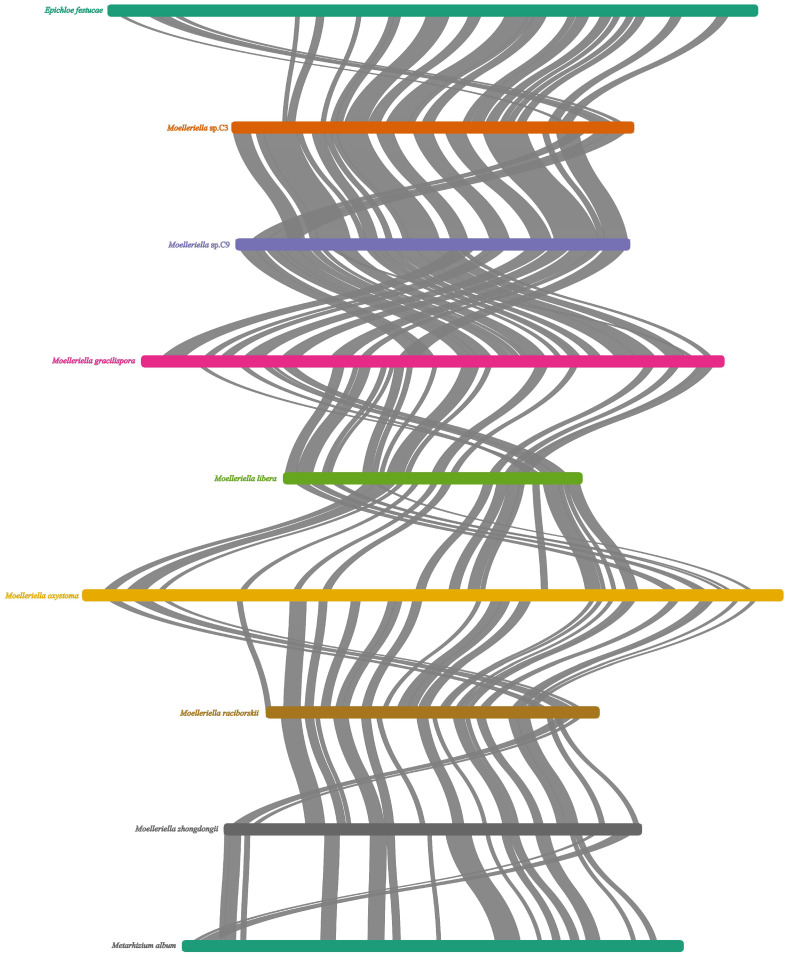
Synteny analysis of seven *Moelleriella* species, with *Metarhizium album* and *Epichloe festucae*. Grey arcs represent homologous regions greater than 500 bp.

## Data Availability

All data generated or analyzed during this study are included in this published article.

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
