# Peer review of "Mitochondrial Genomes from Fungal the Entomopathogenic *Moelleriella* Genus Reveals Evolutionary History, Intron Dynamics and Phylogeny"

_jof, 2025, doi:10.3390/jof11020094_

Round 1
Reviewer 1 Report
Xiong et al. sequenced seven mitogenomes from Moelleriella, which are insect pathogens, and found that extensive intron polymorphism exists among these mitogenomes. They also indicated that the length differences in the mitogenomes among the species depend on intron polymorphism. Using these mitogenomes along with others deposited in NCBI, they demonstrated that this dataset is suitable for molecular phylogenetic analysis, molecular evolutionary analysis of mitochondrial genes, and genome rearrangement analysis of this fungal group. Below are some comments that the authors should address. I hope they will be helpful for the authors to improve the quality of this manuscript.
Major:
P4 2.4. I suggest that the authors show the reason for selecting these 20 NCBI-deposited genomes either here or in the Results section. Are these species representative of each taxonomic group? Additionally, does each of the 20 genomes correspond to those listed in Table S9? If so, please indicated which genomes were selected in Table S9.
Fig 3: Why did the authors use 27 taxa for this analysis? I recommend re-analyzing the data using only the 7 Moelleriella mitogenomes to better highlight the evolutionary patterns specific to Moelleriella.
Fig 4: I suggest that the authors investigated whether introns inserted at the same position are closely related to each other. Understanding this relationship is important for uncovering the dynamics of these introns.
Fig 8: Previous studies, such as Mycoscience 56:66-74(2015) and Studies in mycology 60:1-66 (2008) indicated that Moelleriella is a monophyletic group. However, Fig. 8 shows that Hypocrella discoidea placed within the Moelleriella cluster. Please discuss this incongruence.
Fig 9.: Fig. 9 is unclear. Which parts of the figure represent the synteny blocks? What does “covariance” mean in this context? This term is not explained in Materials and Method section. Please provide a detailed description of the analysis method(s) used and the meaning of the term(s).
I strongly recommend incorporating the different approach to support the statements based on the Fig. 9. For instance, Sci Rep 9:17447 (2019) analyzed sentence using Mauve and clearly identified synteny blocks. Why not applying a similar approach? Additionally, I suggest testing orthologus relationships for the blastn hits listed in Table S10. One way to achieve this is through reciprocal BLAST analysis.
Fig. 9: Hypocrella is used in Fig. 8 and Fig. 9, while not used in Fig 2, 3, 5, 6, 7. Please clarify this inconsistency.
The authors use “mitochondrial genome” and “mitogenome”. Do these phrases mean the same thing or something different?
Please add line numbers to the manuscript to facilitate commenting.
P3 2.1.: Delete the following paragraph: “A total of seven Moelleriella species were used for molecular multilocus nucleotide sequencing. Target loci included the LSU, RPB1 and EF-1α sequences which were amplified and analyzed as described [33-34]. In addition, morphological characteristic for each isolate was determined.” There is no mention of MLST analysis or morphological characterization in the main text.
P4 2.6 Did the authors infer the molecular phylogenetic tree using nucleotide sequences or amino acid sequences? If nucleotide sequences were used, the difference of evolutionary rates among the codon positions can affect tree inference. If a partition model was applied, please describe it in detail.
Fig. 1: The text is difficult to read.
Fig. 1: I suggest that all the mitogenome maps start with the same genes, as shown in Fig 7.
Fig 2A: What does the difference in color represent? Please add it in the figure or describe it in the caption.
Fig.5: the bar for the core PCGFs of Mosp.C3 is too short. Please correct this issue.
P11 3.5. and Fig 6: In the text, the authors use “R”, whereas in Fig. 6, “R2” is used. Please correct them.
In addition, in the text, convert the exponents to superscript. In Fig. 6, convert “e-5“ as “x 10 -5”.
P12 3.6.: The statement “Stachybotrys and Metacordyceps have their nad2 and nad3 genes positioned between the cox1 and atp9 genes rather than between the rps3 and atp9 genes as seen in most Hypocreales mitogenomes analyzed”
does not explain the gene order shown in Fig 7. According the Fig. 7 “Stachybotrys and Metacordyceps have their cox1 gene positioned between the rps1 and nad2 genes rather than between the cab and nad1 genes as seen in most Hypocreales mitogenomes analyzed.” Please correct it.
P13 3.7. What are BPP and BS? Please define these terms.
Fig 8. I could not find the sequence of the Hypocrella discoidea mitogenome in the NCBI database. If the authors sequenced this genome, please indicated it in the text and add the accession number in “2.7. Data availability” section.
Author Response
Dear Editors and Reviewers:
Thank you for your letter and comments relating to our manuscript entitled “Mitochondrial genomes from fungal the entomopathogenic Moelleriella genus reveals evolutionary history, intron dynamics and phylogeny” (ID: jof-3365993). The comments were very helpful in revising and improving our manuscript as well as emphasizing the significance to our research. We have read the comments carefully and made corrections accordingly. Revised portions are marked in blue in the manuscript. The main corrections in the paper and our responses to the reviewer’s comments are given below. We hope that the revisions in the manuscript and our accompanying responses will be sufficient to make our manuscript suitable for publication in the Journal of Fungi.
Responses to the comments of the reviewer:
Reviewer#1
Comments 1: P4 2.4. I suggest that the authors show the reason for selecting these 20 NCBI-deposited genomes either here or in the Results section. Are these species representative of each taxonomic group? Additionally, does each of the 20 genomes correspond to those listed in Table S9? If so, please indicated which genomes were selected in Table S9.
Response 1: The species selected are either the type species of their respective groups or species with high-quality genome assemblies. Additionally, one species was excluded, and the 19 genomes correspond to those listed in Table S9, which have been highlighted in red font in the table.
Comments 2: Fig 3: Why did the authors use 27 taxa for this analysis? I recommend re-analyzing the data using only the 7 Moelleriella mitogenomes to better highlight the evolutionary patterns specific to Moelleriella.
Response 2: The data have been re-analyzed using the mitogenomes of 7 Moelleriella species. The results of our analysis are shown as follows.
Comments 3: Fig 4: I suggest that the authors investigated whether introns inserted at the same position are closely related to each other. Understanding this relationship is important for uncovering the dynamics of these introns.
Response 3: We have expanded the study of the conservation of nucleotide sequences arrounding identical IPSs in Moelleriella species, and the relevant results are presented in the manuscript.
Comments 4: Fig 8: Previous studies, such as Mycoscience 56: 66-74 (2015) and Studies in mycology 60:1-66 (2008) indicated that Moelleriella is a monophyletic group. However, Fig. 8 shows that Hypocrella discoidea placed within the Moelleriella cluster. Please discuss this incongruence.
Response 4: We have revised it.
Comments 5: Fig. 9 is unclear. Which parts of the figure represent the synteny blocks? What does “covariance” mean in this context? This term is not explained in Materials and Method section. Please provide a detailed description of the analysis method(s) used and the meaning of the term(s).
I strongly recommend incorporating the different approach to support the statements based on the Fig. 9. For instance, Sci Rep 9:17447 (2019) analyzed sentence using Mauve and clearly identified synteny blocks. Why not applying a similar approach? Additionally, I suggest testing orthologus relationships for the blastn hits listed in Table S10. One way to achieve this is through reciprocal BLAST analysis.
Response 5: We have revised it. In addition, we further analyzed homologous genes to provide a more comprehensive description of orthologous relationships. The relevant results have also been included in Figure 11, Table S10.
Comments 6: Fig. 9: Hypocrella is used in Fig. 8 and Fig. 9, while not used in Fig 2, 3, 5, 6, 7. Please clarify this inconsistency.
Response 6: We have revised it.
Comments 7: The authors use “mitochondrial genome” and “mitogenome”. Do these phrases mean the same thing or something different?
Response 7: “Mitogenome” is an abbreviation for “mitochondrial genome”, both phrases mean the same thing.
Comments 8: Please add line numbers to the manuscript to facilitate commenting.
Response 8: We have revised it.
Comments 9: P3 2.1.: Delete the following paragraph: “A total of seven Moelleriella species were used for molecular multilocus nucleotide sequencing. Target loci included the LSU, RPB1 and EF-1α sequences which were amplified and analyzed as described [33-34]. In addition, morphological characteristic for each isolate was determined.” There is no mention of MLST analysis or morphological characterization in the main text.
Response 9: We have deleted it.
Comments 10: P4 2.6 Did the authors infer the molecular phylogenetic tree using nucleotide sequences or amino acid sequences? If nucleotide sequences were used, the difference of evolutionary rates among the codon positions can affect tree inference. If a partition model was applied, please describe it in detail.
Response 10: We analyzed the mitogenomes of various species and constructed a phylogenetic tree based on the amino acid sequences.
Comments 11: Fig. 1: The text is difficult to read.
Response 11: We have revised it.
Comments 12: Fig. 1: I suggest that all the mitogenome maps start with the same genes, as shown in Fig 7.
Response 12: We have revised it.
Comments 13: Fig 2A: What does the difference in color represent? Please add it in the figure or describe it in the caption.
Response 13: We have revised it.
Comments 14: Fig.5: the bar for the core PCGFs of Mosp.C3 is too short. Please correct this issue.
Response 14: We have revised it.
Comments 15: P11 3.5. and Fig 6: In the text, the authors use “R”, whereas in Fig. 6, “R2” is used. Please correct them.
In addition, in the text, convert the exponents to superscript. In Fig. 6, convert “e-5“ as “x 10 -5”.
Response 15: We have revised it.
Comments 16: P12 3.6.: The statement “Stachybotrys and Metacordyceps have their nad2 and nad3 genes positioned between the cox1 and atp9 genes rather than between the rps3 and atp9 genes as seen in most Hypocreales mitogenomes analyzed”
does not explain the gene order shown in Fig 7. According the Fig. 7 “Stachybotrys and Metacordyceps have their cox1 gene positioned between the rps1 and nad2 genes rather than between the cab and nad1 genes as seen in most Hypocreales mitogenomes analyzed.” Please correct it.
Response 16: We have revised it.
We compared the arrangements of the 15 core PCGs and 2 rRNAs in the 26 Hypocreales mitogenomes and found that 22 members exhibited identical gene arrangements. However, isolates within Stachybotrys, Metacordyceps, Clonostachys, and Acremonium showed differences and displayed the following gene order: cox1, nad1, nad4, atp8, atp6, rns, cox3, nad6, rnl, rps3, nad2, nad3, atp9, cox2, nad4L, nad5, and cob. The gene orders in the mitogenomes of Acremonium and Clonostachys differed from th others only in the location of the gene cox2, which was found between the nad4 and atp8 genes, whereas Stachybotrys and Metacordyceps had significant differences in gene order, namely as follows: cox1, nad2, nad3, atp9, cox2, nad4L, nad5, cob, nad1, nad4, atp8, atp6, rns, cox3, nad6, rnl and rps3 (Figure 8). In general, the mitogenomes of Hypocreales exhibited a high degree of conservation.
Comments 17: P13 3.7. What are BPP and BS? Please define these terms.
Response 17: We have defined them.
Comments 18: Fig 8. I could not find the sequence of the Hypocrella discoidea mitogenome in the NCBI database. If the authors sequenced this genome, please indicated it in the text and add the accession number in “2.7. Data availability” section.
Response 18: We have revised it.
We tried our best to improve the manuscript and made some changes marked in blue in revised paper which will not influence the content and framework of the paper. We appreciate for Editors/Reviewers’ warm work earnestly and hope the revision will meet with your approval. Once again, thank you very much for your comments and suggestions.
Kind regards,
Junzhi Qiu
E-mail address: junzhiqiu@126.com

Reviewer 2 Report
The manuscript by Xiong et al. focused on mitogenomics of the representatives of the genus Moelleriella is a comprehensive and detailed study that deserves close attention. However, I recommend first thoroughly editing the manuscript and significantly polishing the English, as certain awkward wordings and phrases obstruct clear understanding of the text. Some of them I mentioned below, but far not all.
Recommended corrections to the title of the ms:
Mitochondrial genomes from fungal the entomopathogenic Moelleriella genus reveals evolutionary history, intron dynamics and phylogeny
Recommended corrections to the Abstract:
Lines 1-2 Members of the genus Moelleriella (Hypocreales, Clavicipitaceae) are insect pathogens with specificity for scale insects and whiteflies.
Line 5 varied in size from 40.8 to 95.7 Kb
Lines 7-8 Nevertheless, significant intron polymorphism can be observed among Moelleriella species.
Line 9 ribosomal protein S3 (ribosome protein S3)
Lines 11-13 Comparative analyses of mitogenomes revealed that introns were the primary factor contributing to the size variation in Moelleriella and more broadly within Hypocreales mitogenomes.
Recommended corrections to the Introduction:
Please specify not only the names of taxa but also the taxonomic categories.
Page 2, lines 1-3 - the following editing recommended:
The genus Moelleriella, established by Bresadola in 1896 to accommodate the type species M. sulphurea, belongs to Ascomycota, Sordariomycetes, Hypocreales, Clavicipitaceae.
Page 2, line 22 the “effuse” and “globose” branches
Page 2, line 27 are major significant pests
Page 2, lines 35-37 - recommended correction:
However, a serious challenge is that most species of Moelleriella produce only a limited number of spores in artificial media, making large-scale production difficult.
Recommended corrections to the Materials and methods.
Section 2.2: please, indicate, how many million of raw reads were obtained.
Section 2.4: please, explain the abbreviations when use them for the first time: RSCU - the Relative Synonymous Codon Usage.
Please, correct the spelling: using Condon W.
Please, add to 20 mitogenomes from the NCBI database mentioned here and listed in the Supplementary Table 1 the GenBank Accession Numbers (the numbers better appear in the Table S1, they can be repeated later in the Table S8 as well).
Section 2.6:
Please, use italic for Latin names: Pleurotus ostreatus
Correct the spelling: the outgroup.
‘the PCGs were merged into a combined dataset’ – please, indicate the number of amino acid positions it includes.
Please, make the combined supermatrix available at some public resources like Figshare (https://figshare.com).
‘Newly generated sequences from this study have been deposited in GenBank’. Please, provide accession numbers here if you mention a deposition in the GenBank, one can repeat them below once again in the section ‘Data availability’.
Recommended corrections to the Results:
Page 9, section 3.5, line 2 – misspelling: and where dispersed
Page11, section 3.7, line 1: To reconstruct the evolutionary history lineage of the analysed mitogenomes…
conservation (Fig.8) – missing space
Figure 8, legend, E – use capital letter: (E) Genome sizes of corresponding mitogenomes.
Recommended correction to the Discussion
Page 15, line 14 – please, correct: indicating some intergeneral differences.
Page 15, line 27 – use singular: intron polymorphism
Page 16 – missing spaces: the Aleyrodidae (whiteflies) and Coccidae
Recommended correction to the Supplementary information:
It is a very strange note:
Supplementary data to this article can be found online at GenBank:
https:// www. ncbi. nlm. nih. gov/ genbank/.
Please, provide a reference to the file with the supplementary information.
Title of the Table S8 – spelling mistake should be corrected (ribosomal)
Author Response
Dear Editors and Reviewers:
Thank you for your letter and comments relating to our manuscript entitled “Mitochondrial genomes from fungal the entomopathogenic Moelleriella genus reveals evolutionary history, intron dynamics and phylogeny” (ID: jof-3365993). The comments were very helpful in revising and improving our manuscript as well as emphasizing the significance to our research. We have read the comments carefully and made corrections accordingly. Revised portions are marked in blue in the manuscript. The main corrections in the paper and our responses to the reviewer’s comments are given below. We hope that the revisions in the manuscript and our accompanying responses will be sufficient to make our manuscript suitable for publication in the Journal of Fungi.
Responses to the comments of the reviewer:
Reviewer#2
Comments 1: Recommended corrections to the title of the ms:
Mitochondrial genomes from fungal the entomopathogenic Moelleriella genus reveals evolutionary history, intron dynamics and phylogeny
Response 1: We have revised it.
Comments 2: Recommended corrections to the Abstract:
Lines 1-2 Members of the genus Moelleriella (Hypocreales, Clavicipitaceae) are insect pathogens with specificity for scale insects and whiteflies.
Line 5 varied in size from 40.8 to 95.7 Kb
Lines 7-8 Nevertheless, significant intron polymorphism can be observed among Moelleriella species.
Line 9 ribosomal protein S3 (ribosome protein S3)
Lines 11-13 Comparative analyses of mitogenomes revealed that introns were the primary factor contributing to the size variation in Moelleriella and more broadly within Hypocreales mitogenomes.
Response 2: We have revised it.
Comments 3: Recommended corrections to the Introduction:
Please specify not only the names of taxa but also the taxonomic categories.
Response 3: We have revised it.
The genus Moelleriella, established by Bresadola in 1896 to accommodate the type species M. sulphurea, belongs to Ascomycota, Sordariomycetes, Hypocreales, Clavicipitaceae.
Comments 4: Page 2, lines 1-3 - the following editing recommended:
The genus Moelleriella, established by Bresadola in 1896 to accommodate the type species M. sulphurea, belongs to Ascomycota, Sordariomycetes, Hypocreales, Clavicipitaceae.
Page 2, line 22 the “effuse” and “globose” branches
Page 2, line 27 are major significant pests
Page 2, lines 35-37 - recommended correction:
However, a serious challenge is that most species of Moelleriella produce only a limited number of spores in artificial media, making large-scale production difficult.
Response 4: We have revised them.
Comments 5: Recommended corrections to the Materials and methods.
Section 2.2: please, indicate, how many million of raw reads were obtained.
Response 5: We have revised it.
Comments 6: Section 2.4: please, explain the abbreviations when use them for the first time: RSCU - the Relative Synonymous Codon Usage.
Please, correct the spelling: using Condon W.
Response 6: We have revised it.
Comments 7: Please, add to 20 mitogenomes from the NCBI database mentioned here and listed in the Supplementary Table 1 the GenBank Accession Numbers (the numbers better appear in the Table S1, they can be repeated later in the Table S8 as well).
Response 7: We have revised it.
Comments 8: Section 2.6:
Please, use italic for Latin names: Pleurotus ostreatus
Correct the spelling: the outgroup.
Response 8: We have revised it.
Comments 9: ‘the PCGs were merged into a combined dataset’ – please, indicate the number of amino acid positions it includes.
Response 9: We have revised it.
Comments 10: Please, make the combined supermatrix available at some public resources like Figshare (https://figshare.com).
Response 10: The combined supermatrix has been uploaded to Figshare.
Comments 11:‘Newly generated sequences from this study have been deposited in GenBank’. Please, provide accession numbers here if you mention a deposition in the GenBank, one can repeat them below once again in the section ‘Data availability’.
Response 11: We have revised it.
Comments 12: Recommended corrections to the Results:
Page 9, section 3.5, line 2 – misspelling: and where dispersed
Response 12: We have revised it.
Comments 13: Page11, section 3.7, line 1: To reconstruct the evolutionary history lineage of the analysed mitogenomes…
Response 13: We have revised it.
Comments 14: conservation (Fig.8) – missing space
Response 14: We have revised it.
Comments 15: Figure 8, legend, E – use capital letter: (E) Genome sizes of corresponding mitogenomes.
Response 15: We have revised it.
Comments 16: Recommended correction to the Discussion
Page 15, line 14 – please, correct: indicating some intergeneral differences.
Response 16: We have revised it.
Comments 17: Page 15, line 27 – use singular: intron polymorphism
Response 17: We have revised it.
Comments 18: Page 16 – missing spaces: the Aleyrodidae (whiteflies) and Coccidae
Response 18: We have revised it.
Comments 19: Recommended correction to the Supplementary information:
It is a very strange note:
Supplementary data to this article can be found online at GenBank:
https:// www. ncbi. nlm. nih. gov/ genbank/.
Please, provide a reference to the file with the supplementary information.
Title of the Table S8 – spelling mistake should be corrected (ribosomal)
Response 19: We have revised them.
We tried our best to improve the manuscript and made some changes marked in blue in revised paper which will not influence the content and framework of the paper. We appreciate for Editors/Reviewers’ warm work earnestly and hope the revision will meet with your approval. Once again, thank you very much for your comments and suggestions.
Kind regards,
Junzhi Qiu
E-mail address: junzhiqiu@126.com

Round 2
Reviewer 1 Report
All of the comments have been appropriately addressed, except for one concerning Figure 10.
The caption in Fig 10 states that the red arcs indicate ‘inverted’ regions. There are lots of red lines observed between Moelleriella sp. C9 and Moelleriella sp. C3, for example. However, in the Tables S9, no inversion sequences were observed in the blastne results between Moelleriella sp. C9 (query) and Moelleriella sp. C3 (database). All the blastn results indicated that the homologous regions are the same directions (d.start < d.end). Please clalify this issue or I recommend deleting Fig 10. I think that the red arcs indicate not inverted regions, but something like edges of co-linear fragments.
There are no commnets for here.
Author Response
Dear Editors and Reviewers:
Thank you for your letter and comments relating to our manuscript entitled “Mitochondrial genomes from fungal the entomopathogenic Moelleriella genus reveals evolutionary history, intron dynamics and phylogeny” (ID: jof-3365993). The comments were very helpful in revising and improving our manuscript as well as emphasizingthe significance to our research. We have read the comments carefully and made corrections accordingly. Revised portions are marked in blue in the manuscript. The main corrections in the paper and our responses to the reviewer’s comments are given below. We hope that the revisions in the manuscript and our accompanying responses will be sufficient to make our manuscript suitable for publication in the Journal of Fungi.
Responses to the comments of the reviewer:
Reviewer#1
Comments 1: The caption in Fig 10 states that the red arcs indicate‘inverted’ regions. There are lots of red lines observed between Moelleriella sp. C9 and Moelleriella sp. C3, for example. However, in the Tables S9, no inversion sequences were observed in the blastne results between Moelleriella sp. C9 (query) and Moelleriella sp. C3 (database). All the blastn results indicated that the homologous regions are the same directions (d.start < d.end). Please clarify this issue or I recommend deleting Fig 10. I think that the red arcs indicate not inverted regions, but something like edges of co-linear fragments.
Response 1: We have deleted Figure 10.
We tried our best to improve the manuscript and made some changes marked in blue in revised paper which will not influence the content and framework of the paper. We appreciate for Editors/Reviewers’ warm work earnestly and hope the revision will meet with your approval. Once again, thank you very much for your comments and suggestions.
Kind regards,
Junzhi Qiu
E-mail address: junzhiqiu@126.com
